# Long-context LLMs Struggle with Long In-context Learning

♠,♣**Tianle Li,** ♠,♣**Ge Zhang,** ♠**Quy Duc Do,** †**Xiang Yue,** ♠,♣**Wenhu Chen**
♠**University of Waterloo**
†**Carnegie Mellon University**
♣**Vector Institute, Toronto**
`{t29li,wenhuchen}@uwaterloo.ca`
`https://github.com/TIGER-AI-Lab/LongICLBench`

Reviewed on OpenReview: `https://openreview.net/forum?id=Cw2xlg0e46`

## Abstract

Large Language Models (LLMs) have made significant strides in handling long sequences. Some models like Gemini could even be capable of dealing with millions of tokens. However, their performance evaluation has largely been confined to metrics like perplexity and synthetic tasks, which may not fully capture their true abilities in more challenging, real-world scenarios. We introduce a benchmark (*LongICLBench*) for long in-context learning in extreme-label classification using six datasets with 28 to 174 classes and input lengths from 2K to 50K tokens. Our benchmark requires LLMs to comprehend the entire input to recognize the massive label spaces to make correct predictions. We evaluate on 15 long-context LLMs and find that they perform well on less challenging classification tasks with smaller label space and shorter demonstrations. However, they struggle with more challenging task like Discovery with 174 labels, suggesting a gap in their ability to process long, context-rich sequences. Further analysis reveals a bias towards labels presented later in the sequence and a need for improved reasoning over multiple pieces of information. Our study reveals that long context understanding and reasoning is still a challenging task for the existing LLMs. We believe *LongICLBench* could serve as a more realistic evaluation for the future long-context LLMs.

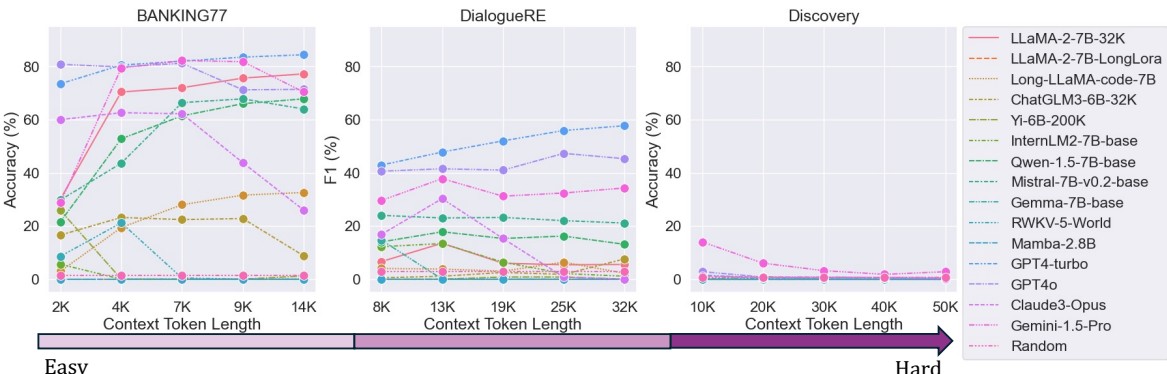

Figure 1: LLM performance on long in-context benchmark across different lengths. We curate datasets with different difficulty levels. As we increase the difficulty of the dataset, LLMs struggle to understand the task definition and suffer from significant performance degradation.

# 1 Introduction

Large language models have already entered the long context era. A myriad of LLMs has been released to support long context windows from 32K to 2M tokens. These methods (Hao et al., 2022; Chen et al., 2023a; Peng et al., 2023b; Ratner et al., 2023; Xiao et al., 2024; Jin et al., 2024) can unlock lots of complex real-world applications, such as long-document question-answering, multi-document summarization, long-horizon agent tasks, and repo-level code understanding.

One line of research is based on AliBi (Press et al., 2022) and RoPE (Su et al., 2024) embeddings, which allows us to train Transformers with short sequences and subsequently apply them to longer sequences during inference. Recently, different approaches (Xiong et al., 2023; Fu et al., 2024; Liu et al., 2024) help the model to extrapolate to 128K window size with continued pre-training. Later on, LongRoPE (Ding et al., 2024) was proposed to further extend the context window to 2M tokens. Another line of research also utilizes methodologies like context window sliding and segmentation to overcome the issue of the limited context window in original Transformers (Hao et al., 2022; Ratner et al., 2023). Furthermore, architectural innovations, transitioning from traditional Transformer-based designs to recurrent models or state space models, have shown promise in facilitating long-range computations naturally (Orvieto et al., 2023; Gu & Dao, 2023; Peng et al., 2023a). These techniques have been incorporated into several current open-source LLMs to enhance long sequence understanding capability (Chen et al., 2023b; Tworkowski et al., 2023).

These long-context models are primarily evaluated on three types of evaluations:
1. language model perplexity over long documents, which is used by most papers.
2. passkey retrieval (Mohtashami & Jaggi, 2023; Chen et al., 2023a; Li et al., 2023a) or needle-in-a-haystack (Team et al., 2023; Fu et al., 2024), which requires reciting a randomly inserted information in a long sequence. Several LLMs achieve 99%+ on this synthetic task.
3. long-document question-answer or summarization over Qasper (Dasigi et al., 2021).

Evaluations (1) and (2) only provide a minimum bar for LLMs to pass, but their results cannot reflect LLMs' true ability to deal with realistic long-sequence tasks. Evaluation (3) provides a more realistic metric,

There is an important info hidden inside a lot of irrelevant text. Find it and memorize them. I will quiz you about the important information there. <prefix filler by continuously repeating: The grass is green. The sky is blue. The sun is yellow. Here we go. There and back again.> The pass key is <PASS KEY>. Remember it. <PASS KEY> is the pass key. <suffix filler> What is the pass key? The pass key is

**(a) Passkey Retrieval**

**Passage:** Mark Hunter (Slater), a high school student in a sleepy suburb of Phoenix, Arizona, starts an FM pirate radio station that broadcasts from the basement of his parents' house. Mark is a loner, an outsider……
**Question:** Who is Mark Hunter?
**Answer:** He is a high school student in Phoenix

**(b) Long-document Question-answer**

**In-context Prompt:**
**Dialogue:** Speaker 1: Hey Rach, can I talk to you outside for a second? …… Speaker 1: Ah well, can't blame a guy for trying!",
**Predict the relationship between the following entity pairs:**
Speaker 2 and Rach, Speaker 2 and Phoebe, Speaker 2 and Speaker 1
**Answer:** per:alternate_names, per:roommate, per:girl/boyfriend
…
**Dialogue:** Speaker 1: You've got to get back out there, it's your party……\nSpeaker 3: Didn't you like, just get your eyes checked?\nSpeaker 1: Well yeah, but, you know, uh, 27 is a dangerous eye age.",
**Predict the relationship between the following entity pairs:**
Speaker1 and 27, Speaker1 and Speaker 2, Speaker 2 and opthamologists
**Answer:** per:age, per:positive_impression, per:title
…
**Dialogue:** Speaker 1: Hello?\nSpeaker 2: Joey just called. He's got courtside Knicks tickets for him and me tomorrow night. …… \nSpeaker 2: Yeah, ah, ah.. . I'll think of something."
**Predict the relationship between the following entity pairs:**
Speaker 1 and Speaker 2, Joe and Knicks, Speaker 1 and restaurant
**Answer:** per:spouse, per:positive_impression, per:place_of_work

**(c) Extreme-label In-context Learning**

Figure 2: Comparison extreme-label ICL with existing evaluation tasks. Passkey Retrieval is a synthetic task. Long-document Question-answering does not require reading the entire document to find the answer. In extreme-label ICL, the model needs to scan through the entire demonstration to understand the whole label space to make the correct prediction.

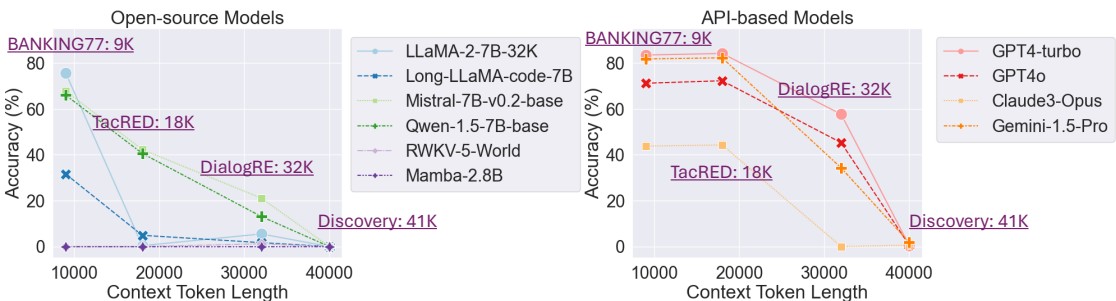

Figure 3: Results for representative models across different evaluation datasets. The performance greatly decreases as the task becomes more challenging.

however, these tasks are more focused on retrieving correct information from the long input. In question answering, LLMs can take a shortcut to read a short snippet to predict the answer without reading the entire document as demonstrated in Figure 2 case (b). Similarly, summarization also suffers from the strong position bias, where LLMs can utilize the few leading sentences (Nallapati et al., 2017) to achieve high performance. Therefore, these metrics are insufficient to measure LLMs' ability to comprehend and reason over the entire input sequence.

In this paper, we propose to adopt in-context learning (ICL) on extreme-label classification tasks (Anil et al., 2022; Milios et al., 2023) to evaluate long-context LLMs. Unlike the prior tasks, in-context learning requires LLMs to recognize the task by scanning over the entire input to understand the label space. This task necessitates LLMs' ability to comprehend the entire input to make predictions. Due to the massive label space, the task demonstration could easily become a long sequence. For example, Discovery (Sileo et al., 2019) encompasses 174 classes with each example taking an average of 61 tokens. Therefore, the minimum total demonstration length (1 shot per class) already exceeds 10K tokens. Normally, LLMs demand more than 1 shot per class to understand the nuances of different fine-grained labels. Having multiple shots can significantly extend the total demonstration length to above 32K. Therefore, this task becomes a natural testbed for long-context understanding.

To systematically assess how these extended input capabilities affect model performance in the realm of fine-grained text classification with in-context learning, we have compiled a benchmark, i.e. *LongICLBench*, consisting of six carefully-selected tasks with different difficulty levels in terms of context length and label space. We evaluate the performance of a wide range of long-context LLMs and **find that the performance of the open-source models uniformly dips as the task becomes more complex (e.g. requiring longer demonstration) as shown in Figure 3. Among the open-source models, the non-Transformer-based models, like RWKV and Mamba (Peng et al., 2023a; Gu & Dao, 2023), perform far behind the Transformer-based models. Simultaneously, within a task, most of the models can benefit from the extensive demonstration if the length is within a certain range. As the input grows longer, it either hurts or makes the performance fluctuate as shown in Figure 1**.

On the most difficult extreme-label classification task Discovery (Sileo et al., 2019), all LLMs achieve close-to-zero performance except Gemini-1.5-Pro with 14% accuracy. In contrast, a fine-tuned BERT model (Kenton & Toutanova, 2019) can achieve 87%. This highlights the challenges that the long in-context learning pose for the existing LLMs. Moreover, we make further analysis on the distribution of label position to investigate the factors that affect the long in-context learning capability of these models. It is shown that the position distribution of instances in the prompt can dramatically influence the performance of some of the evaluated models.

In a nutshell, our contributions to this work can be summarized as follows:

1. We have identified in-context learning on extreme-label classification tasks as an ideal testbed for the evaluation of the long-context capability of the current LLMs. We developed *LongICLBench*, which serves as a complement to earlier benchmarks that concentrated on tasks like long document summarization, question answering (QA), or retrieval, focusing instead on long in-context learning.

2. We evaluate a line of recent long-context LLMs on *LongICLBench* and reveal their performances with gradually changed difficulty levels. Simultaneously, we find the sensitivity of some of the long-context LLMs regarding instance position in the prompt. We hope the evaluation results can provide more insights for the improvement of the design of long-context large language models.

## 2 Related Work

**Long In-context Learning on LLMs** As pre-trained language models continue to grow in size, in-context learning (ICL) has emerged as a favored approach for addressing a wide array of tasks without the need for extensive fine-tuning (Dong et al., 2023). A body of research has established that increasing the number of examples demonstrations can enhance ICL performance (Liu et al., 2022; Wu et al., 2023). Nonetheless, there are studies indicating that longer input prompts can actually diminish performance (Liu et al., 2023), with the effectiveness of prior large language models (LLMs) being constrained by the maximum sequence length encountered during their training. It is also claimed in previous works that LLM+ICL falls short on specification-heavy tasks due to inadequate long-text understanding ability (Peng et al., 2023c). To counter this issue, various works have introduced memory augmentation and extrapolation techniques to support ICL with an extensive set of demonstrations (Li et al., 2023c; Wang et al., 2023).

**Long Context Techniques over LLMs** The effectiveness of Transformer-based models is hindered by the quadratic increase in computational cost relative to sequence length, particularly in handling long context inputs. Recent efforts have explored various strategies to address this challenge. Some studies have pursued continued fine-tuning of the LLM with longer context inputs (Rozière et al., 2024; Tworkowski et al., 2023). Others have leveraged position extrapolation or interpolation, building upon relative rotary positional embedding (Su et al., 2021), to extend input length beyond the training phase (Press et al., 2022; Chen et al., 2023a). Additionally, more approaches have been proposed to mitigate computational issues, including sliding memory window and chunk segmentation (Hao et al., 2022; Ratner et al., 2023; Zhu et al., 2024). Furthermore, alternative architectures beyond Transformer have been explored to handle long inputs more naturally, such as selective-state-spaces models (Peng et al., 2023a; Gu & Dao, 2023). These diverse approaches claim that they can enhance the capabilities of LLMs in processing long context inputs more efficiently.

**Long Context Evaluation** Due to the imperious demands for the support of long-range LLMs, there is a series of benchmarks focusing on long context evaluation. Long-Range Arena (Tay et al., 2021) includes tasks consisting of sequences ranging from 1K to 16K tokens to evaluate variations of fast Transformers. LongBench (Bai et al., 2023b) comprises 21 bilingual datasets with an average length of around 6k words, which have been processed in a unified format to enable effortless evaluation. L-Eval Benchmark (An et al., 2023) supports 20 sub-tasks with input lengths of 3K to 200K tokens. LooGLE (Li et al., 2023b) focuses on summarization and long dependency QA tasks with test instances exceeding 100k words. Most recently, ∞Bench (Zhang et al., 2024) encompasses 12 tasks with an average length of 200K tokens. Another recent work explores the impact of extending input lengths, especially on reasoning tasks (Levy et al., 2024).

**Extreme-label Classification** Extreme-label Classification involves categorizing data into one of an extremely large number of labels, and finds application across a variety of real-world domains such as emotion classification, named entity recognition, and biological function prediction, each requiring precise differentiation among vast label spaces (Zhang et al., 2017; Sileo et al., 2019; Demszky et al., 2020; Ding et al., 2021). Previous methods to tackle Extreme-label Classification tasks range from embedding-based approaches to fine-tuned retrievals (Bhatia et al., 2015; Vulić et al., 2021).However, integrating this task with long-context large language models presents unique challenges. The large scale of the label space complicates the in-context learning process, where LLMs are expected to discern fine-grained differences among labels based on extensive context (Milios et al., 2023). These challenges make the proposed *LongICLBench* with a range of difficulty levels a good testing scenario to evaluate the capability of long-context large language models.

Table 1: Statistics of the collected sub-dataset in *LongICLBench*. We evaluate from 1-shot/label to 5-shot/label, which results in the shown #total token range.

| Dataset | Task Type | # Classes | # Tokens/Shot | # Total Tokens |
|---------|-----------|-----------|---------------|----------------|
| GoEmotion | Emotion Classification | 28 | 28 | [1K, 4K] |
| BANKING77 | Intent Classification | 77 | 28 | [2K, 11K] |
| TacRED | Relation Extraction | 41 | 80 | [4K, 18K] |
| Few-NERD | Entity Recognition | 66 | 61 | [5K, 23K] |
| DialogRE | Relation Extraction | 36 | 226 | [8K, 32K] |
| Discovery | Discourse Marker Classification | 174 | 61 | [10K, 50K] |

## 3 Long In-context Evaluation

### 3.1 Long In-context Benchmark

To support the evaluation of long in-context learning on extreme-label classification tasks in different domains and various difficulty levels, we collect six datasets containing context length from short to long. In order to balance the sequence token length within each dataset and the goal of evaluation for long in-context learning, we keep a subset of the classes among all the classes to format evaluation sets around 1 round, 2 rounds, 3 rounds, 4 rounds, and 5 rounds correspondingly, where each round represent a complete set of examples containing all unique chosen labels. We sample the number of instances from each of the classes evenly to reduce the bias resulting from the label distribution. The statistics of the datasets are described in detail in Table 1.

**GoEmotions** (Demszky et al., 2020) is the largest manually annotated dataset of 58k English comments from Reddit, which is labeled into 27 emotion categories or Neutral. Each selected example contains 28 tokens on average.

**BANKING77** (Casanueva et al., 2020) is a banking-domain intent detection dataset comprising 13,083 annotated examples over 77 intents. We keep all of the types of intents, and each of the instances contains around 28 tokens.

**TacRED** (Zhang et al., 2017) is a large-scale relation extraction dataset with 106,264 examples built over news and web text. Only one relation is labeled for each of the sentences in the dataset. It covers 41 relation types in total, with an average length of 80 tokens for each example.

**Few-NERD** (Ding et al., 2021) is a human-annotated name entity recognition dataset with a hierarchy of 8 coarse-grained and 66 fine-grained entity types. Each of the instances is a paragraph with about 61 tokens on average and contains one or multiple entity names as the ground truth answer.

**DialogRE** (Yu et al., 2020) is a human-annotated dialogue-based relation extraction dataset from an American television comedy, Friends, with 36 possible relation types existing between an argument pair in a dialogue. Each example contains an average of 226 tokens.

**Discovery** (Sileo et al., 2019) automatically discovers sentence pairs with relevant discourse markers and forms a dataset containing 174 discourse markers with at least 10K examples each. Each example contains around 61 tokens. This dataset is the most difficult task with fine-grained labels.

### 3.2 Model and Experimental Setup

In the exploration of in-context learning for extreme-label classification, we conduct a comprehensive evaluation of popular open-source long-context language models of size around 7B parameters. We also include SoTA models like Gemini-1.5-Pro, Claude3-Opus, and GPT-4-turbo. Table 2 provides an overview of the models investigated, highlighting the innovations in their architecture specifically for dealing with long context. We can observe that there are multiple strategies adopted to extend the context window. Some of the models support the training context window size while some models support length extrapolation. RWKV (Peng

et al., 2023a) and Mamba (Gu & Dao, 2023) are the two new RNN-like architectures to decrease attention complexity, which would allow the model to easily extrapolate to much longer inputs with linear time/memory complexity.

Table 2: The overview of the evaluated models. We utilize base models before instruction-tuning except API-based models. LF means fine-tuning the model on longer-context corpus after pre-training.

| Model | Size | Initialization | Strategy | Train | Support |
|---|---|---|---|---|---|
| Gemma-7B-base | 7B | Gemma | RoPE + LF | 8K | 8K |
| LLaMA-2-7B-32K | 7B | LLaMA-2 | Position Interpolation | 32K | 32K |
| ChatGLM3-6B-32K | 6B | ChatGLM | Position Encoding Scheme | 32K | 32K |
| Qwen-1.5-7B-base | 7B | Qwen | NTK-Aware Interpolation | 32K | 32K |
| Mistral-7B-v0.2-base | 7B | Mistral | LF | 32K | 32K |
| LLaMA-2-7B-LongLora | 7B | LLaMA-2 | Shifted Short Attention | 100K | 100K |
| Yi-6B-200K | 6B | Yi | Position Interpolation +LF | 200K | 200K |
| InternLM2-7B-base | 7B | InternLM | Dynamic NTK | 32K | 200K |
| Long-LLaMA-code-7B | 7B | LLaMA-2 | Focused Transformer | 8K | 256K |
| RWKV-5-World | 3B | RWKV | Attention-free Model | 4K | $\infty$ |
| Mamba-2.8B | 2.8B | Mamba | State Space Model | 2K | $\infty$ |
| GPT4-turbo | - | GPT-4 | - | - | 128K |
| GPT4o | - | GPT-4 | - | - | 128K |
| Claude3-Opus | - | Claude3 | - | - | 200K |
| Gemini-1.5-Pro | - | Gemini | - | - | 10M |

We construct a prompt following the template as shown in A.2 for each of the datasets. To fairly evaluate the open-source and API-based models with a series of input lengths, we sample the same example set for all the models with labels distributed evenly to ensure an unbiased distribution for the in-context demonstration. For instance, an input of one round will include one set of examples traversing all the types, and 5 rounds will contain instances from each of the labels 5 times. For testing, we sample 500 examples from the test set of each dataset, simultaneously ensuring an even distribution in terms of the type of labels. All the open-source models are loaded from the weights in HuggingFace[1], and inferred on eight NVIDIA RTX A6000 GPUs, while the API-based models are based on the official documentations [2].

### 3.3 Experiment Result

The main evaluation results are demonstrated in Table 3, Table 4, Table 5, Table 6 and subsection A.1. For the entity recognition and relationship extraction dataset, we use the F1 score as the evaluation metric, and Accuracy is utilized for the other datasets. From the presented results, generally, we can find that models of Transformer-based architecture perform consistently better than the RNN-based ones in all the evaluated datasets. However, both of them are still falling behind the powerful API-based models. For a relatively simple task like BANKING77, whose context length from 1 round to 5 rounds is 2K to 14 K, most of the models can benefit from the extensive context with more demonstrations. As shown in Figure 1 and Table 3, from 2K to 4K, there is either a huge increase nearly doubling the accuracy, or a complete failure for most of the open-source models. After 3 rounds, limited performance gain can be achieved by adding more examples. When it comes to more complicated tasks like TacRED and DialogueRE in Table 4 and Table 5, which are more urgently requiring the capability of long-context comprehension, the overall performance of all the few-shot models drops compared to BANKING77. As shown in the middle plot of Figure 1, only GPT4-turbo and GPT4o can consistently benefit from more demonstrations, all of the other models reach their peak at the middle with context length around 13K to 25K.

For the most challenging Discovery dataset, which has an extremely large label space including 174 classes, one round of traversing for all the label possibilities has already made up a context length of 10K. In this

---

[1] https://huggingface.co
[2] https://platform.openai.com/docs/guides/text-generation/chat-completions-api, https://cloud.google.com/vertex-ai/generative-ai/docs/multimodal/overview

extreme case, all of the models except Gemini-1.5-Pro, fail to tell the difference among the fine-grained types including GPT4-turbo, leading to a score of 0. The results across different datasets reveal the models' capability to understand different types of tasks. Our initial hypothesis suggests that the strongest LLMs like GPT-4-turbo are capped at a certain complexity level between DialogRE and Discovery.

Another interesting observation we have is that some LLMs' performance on the extreme-label ICL seems highly predictable. According to the left sub-graph of Figure 3, the performance of Qwen and Mistral is almost linear w.r.t the demonstration length. This reveals that there might be an underlying mathematical relation between performance and the task complexity for ICL.

Table 3: BANKING77 result with respect to increasing context length. **1R** represents one round of traversing all the instances with a unique label.

| Model | Param | Support | 1R | 2R | 3R | 4R | 5R |
|---|---|---|---|---|---|---|---|
| **Context Tokens** | | | **2K** | **4K** | **7K** | **9K** | **14K** |
| Gemma-7B-base | 7B | 8K | 0 | 0 | 0 | 0 | 0 |
| LLaMA-2-7B-32K | 7B | 32K | **30.2** | **70.4** | **72.0** | **75.6** | **77.2** |
| ChatGLM3-6B-32K | 6B | 32K | 16.6 | 23.2 | 22.4 | 22.8 | 8.8 |
| Qwen-1.5-7B-base | 7B | 32K | 21.6 | 52.8 | 61.4 | 66.0 | 67.8 |
| Mistral-7B-v0.2-base | 7B | 32K | 29.8 | 43.6 | 66.4 | 67.8 | 64.0 |
| LLaMA-2-7B-LongLora | 7B | 100K | 0 | 0 | 0 | 0 | 0 |
| Yi-6B-200K | 6B | 200K | 25.8 | 0 | 0 | 0 | 1.2 |
| InternLM2-7B-base | 7B | 200K | 5.6 | 0 | 0 | 0 | 0 |
| Long-LLaMA-code-7B | 7B | 256K | 3.0 | 19.4 | 28.0 | 31.6 | 32.6 |
| RWKV-5-World | 7B | 4K | 8.6 | 21.2 | 0.4 | 0 | 0 |
| Mamba-2.8B | 2.8B | 2K | 0 | 0 | 0 | 0 | 0 |
| GPT4-turbo | N/A | 128K | **73.5** | **80.5** | 82.0 | **83.5** | **84.4** |
| GPT4o | N/A | 128K | 80.8 | 79.8 | 81.2 | 71.2 | 71.4 |
| Claude3-Opus | N/A | 200K | 60.0 | 62.6 | 62.2 | 43.8 | 26.0 |
| Gemini-1.5-Pro | N/A | 10M | 28.8 | 79.4 | **82.2** | 81.8 | 70.4 |
| SoTA (RoBERTA + ICDA) | N/A | - | | | **94.4** | | |

Table 4: TacRED result with respect to increasing context length.

| Model | Param | Support | 1R | 2R | 3R | 4R | 5R |
|---|---|---|---|---|---|---|---|
| **Context Tokens** | | | **4K** | **7K** | **10K** | **14K** | **18K** |
| Gemma-7B-base | 7B | 8K | 0.4 | 0.4 | 0 | 0 | 0 |
| LLaMA-2-7B-32K | 7B | 32K | 0 | 0.4 | 0.4 | 0.8 | 0.4 |
| ChatGLM3-6B-32K | 6B | 32K | 29.7 | 36.1 | 38.9 | 40.1 | 25.2 |
| Qwen-1.5-7B-base | 7B | 32K | 38.7 | 47.3 | 45.2 | 43.6 | 40.6 |
| Mistral-7B-v0.2-base | 7B | 32K | **53.3** | **53.1** | **51.6** | **48.0** | **42.3** |
| LLaMA-2-7B-LongLora | 7B | 100K | 0 | 0 | 0 | 0 | 0 |
| Yi-6B-200K | 6B | 200K | 5.6 | 1.9 | 8.0 | 9.5 | 2.0 |
| InternLM2-7B-base | 7B | 200K | 29.6 | 27.2 | 15.5 | 10.7 | 8.0 |
| Long-LLaMA-code-7B | 7B | 256K | 3.8 | 7.1 | 4.1 | 6.6 | 4.9 |
| RWKV-5-World | 7B | 1K | 2.3 | 2.6 | 1.0 | 0 | 1.2 |
| Mamba-2.8B | 2.8B | 2K | 0 | 0 | 0 | 0 | 0 |
| GPT4-turbo | N/A | 128K | **74.4** | 76.5 | 79.5 | 80.4 | **84.2** |
| GPT4o | N/A | 128K | 71.1 | 75.5 | 73.6 | 73.2 | 72.3 |
| Claude3-Opus | N/A | 200K | 68.7 | 74.1 | 35.4 | 43.4 | 44.3 |
| Gemini-1.5-Pro | N/A | 10M | 72.6 | **81.4** | **79.6** | **81.4** | 82.3 |
| SoTA (DeepStruct) | N/A | - | | | 76.8 | | |

Table 5: DialogRE result with respect to increasing context length.

| Model | Param | Support | 1R | 2R | 3R | 4R | 5R |
|---|---|---|---|---|---|---|---|
| **Context Tokens** | | | **8K** | **13K** | **19K** | **25K** | **32K** |
| Gemma-7B-base | 7B | 8K | 14.7 | 0 | 0 | 0 | 0 |
| LLaMA-2-7B-32K | 7B | 32K | 6.6 | 13.5 | 6.0 | 5.4 | 5.5 |
| ChatGLM3-6B-32K | 6B | 32K | 0.5 | 1.1 | 2.5 | 1.8 | 7.6 |
| Qwen-1.5-7B-base | 7B | 32K | 14.0 | 17.8 | 15.3 | 16.2 | 13.1 |
| Mistral-7B-v0.2-base | 7B | 32K | **24.0** | **23.0** | **23.2** | **22.0** | **21.1** |
| LLaMA-2-7B-LongLora | 7B | 100K | 0 | 0 | 0 | 0 | 0 |
| Yi-6B-200K | 6B | 200K | 0 | 0 | 0.4 | 0.4 | 0 |
| InternLM2-7B-base | 7B | 200K | 12.0 | 13.2 | 5.8 | 1.8 | 0.7 |
| Long-LLaMA-code-7B | 7B | 256K | 2.7 | 3.0 | 2.6 | 5.2 | 1.7 |
| RWKV-5-World | 7B | 4K | 0 | 0 | 0 | 0 | 0 |
| Mamba-2.8B | 2.8B | 2K | 0 | 0 | 0 | 0 | 0 |
| GPT4-turbo | N/A | 128K | **42.9** | **47.8** | **52.0** | **55.9** | **57.7** |
| GPT4o | N/A | 128K | 40.6 | 41.5 | 41.0 | 47.3 | 45.3 |
| Claude3-Opus | N/A | 200K | 16.8 | 30.3 | 15.3 | 0.8 | 0 |
| Gemini-1.5-Pro | N/A | 10M | 29.6 | 37.8 | 31.2 | 32.4 | 34.3 |
| SoTA (HiDialog) | N/A | - | | | **77.1** | | |

Table 6: Discovery result with respect to increasing context length.

| Model | Param | Support | 1R | 2R | 3R | 4R | 5R |
|---|---|---|---|---|---|---|---|
| **Context Tokens** | | | **10K** | **20K** | **30K** | **40K** | **50K** |
| Gemma-7B-base | 7B | 8K | 0 | 0 | 0 | 0 | 0 |
| LLaMA-2-7B-32K | 7B | 32K | 0 | 0 | 0 | 0 | ✗ |
| ChatGLM3-6B-32K | 6B | 32k | 0 | 1.0 | 0 | ✗ | ✗ |
| Qwen-1.5-7B-base | 7B | 32K | 0 | 0 | 0 | 0 | 0 |
| Mistral-7B-v0.2-base | 7B | 32K | 0 | 0 | 0 | 0 | 0 |
| LLaMA-2-7B-LongLora | 7B | 100K | 0 | 0 | 0 | 0 | 0 |
| Yi-6B-200K | 6B | 200k | 0 | 0 | 0 | 0 | 0 |
| InternLM2-7B-base | 7B | 200K | 0 | 0 | 0 | 0 | 0 |
| Long-LLaMA-code-7B | 7B | 256K | 0 | 0 | 0 | 0 | 0 |
| RWKV-5-World | 7B | 4K | 0 | 0.2 | 0 | 0 | 0 |
| Mamba-2.8B | 2.8B | 2K | 0 | 0 | 0 | 0 | 0 |
| GPT4-turbo | N/A | 128K | 1.5 | 0.5 | 0.5 | 0.5 | 0.5 |
| GPT4o | N/A | 128K | 2.8 | 0.8 | 0.8 | 0.6 | 0.4 |
| Claude3-Opus | N/A | 200K | 1.2 | 0.6 | 0.6 | 0.6 | 0.2 |
| Gemini-1.5-Pro | N/A | 10M | **14.0** | **6.0** | **3.2** | **1.8** | **2.8** |
| SoTA (MTL) | N/A | - | | | **87.4** | | |

# 4 Exploratory Experiment

Inspired by the Lost in the Middle phenomenon (Liu et al., 2023), we take analysis experiments to explore whether the position distribution of the instances will make a difference in the performance for long in-context learning with extreme-label classification tasks.

## 4.1 Scattered Distribution

In our investigation, we conduct pilot experiments on TacRED, a medium-complexity dataset, with each label type demonstrated three times, resulting in a total of 123 distinct instances (calculated as $41 \times 3$).

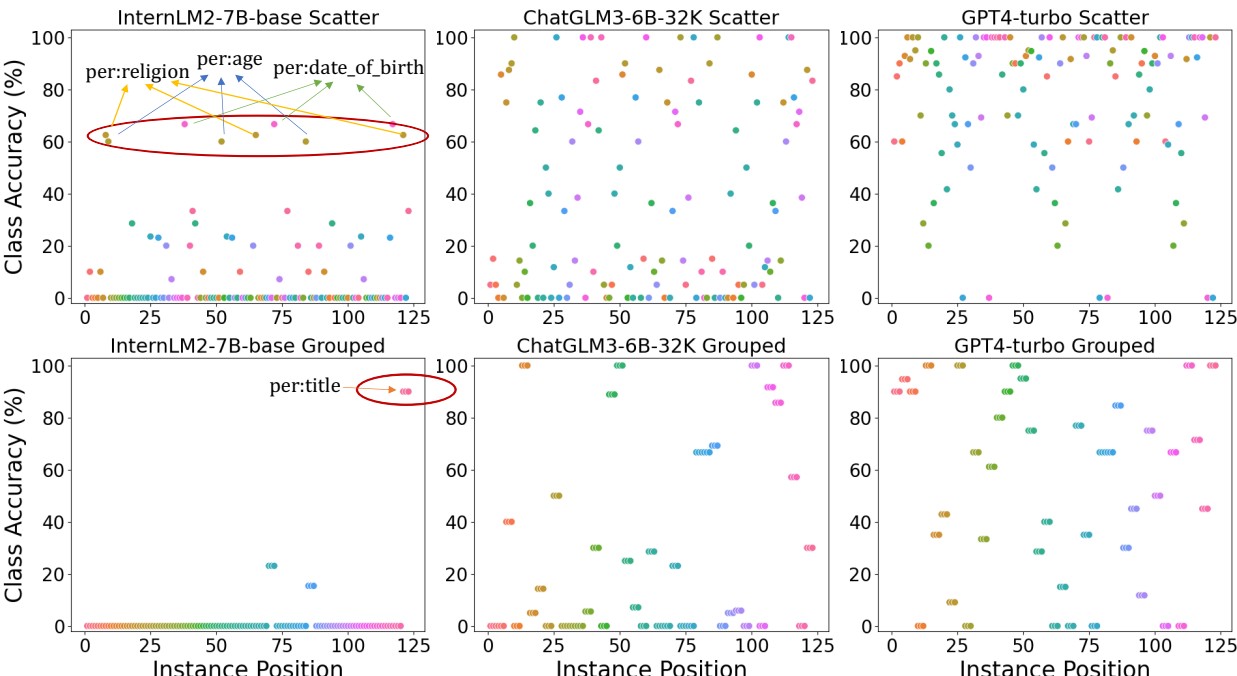

Figure 4: Visualization of accuracy for every class when instances from the same class are scattered V.S. grouped in the demonstration prompt.

Within these experiments, instances bearing the same labels are distributed randomly to form a scattered configuration. For each instance, we track its relative position within the prompt alongside its corresponding label, thereafter computing the accuracy for each label class. As illustrated in the first row of Figure 4, the visualization delineates the accuracy of each label, aligned with its position within the prompt, where diverse colors symbolize various label types. In scenarios where class instances are scattered, certain models, such as InternLM2-7B-base, demonstrate acceptable performances—approximately 60% accuracy merely on specific labels, as highlighted by a red circle in Figure 4, regardless of the instance placements. Conversely, other models, like ChatGLM3-6B-32K, exhibit robust performance across a broad spectrum of labels. Remarkably, the GPT4-turbo model consistently surpasses an 80% accuracy threshold for the majority of label types, with only a minimal count of exceptions.

## 4.2 Grouped Distribution

To facilitate a clear comparison between scattered and grouped distributions, we organize instances of the same class to be adjacent within the demonstration prompts. The impact of this reorganization on model performance, both pre and post-grouping, is presented in subsection A.3. It is easy to observe that there is a general decline in performance across most models after grouping instances by class. Notably, models such as Mistral and InternLM2 exhibit significant performance drops, underscoring a pronounced sensitivity to instance grouping. In an effort to delve deeper into this phenomenon, we visualize the accuracy of grouped labels in relation to their positions within the prompt, as illustrated in Figure 4. This visualization reveals that instances of the same class, denoted by dots of the same color, are positioned nearby. It became evident that some models, like InternLM2 or Mistral shown in subsection A.3, demonstrate high sensitivity to the distribution of instances, only handling instances with labels positioned at the end of the prompt. Conversely, other open-source models such as ChatGLM3-6B-32K, with a modest 3.3% drop in accuracy, proved to be more resilient to changes in instance positioning. Surprisingly, even the GPT4-turbo and Gemini1.5-Pro are not immune to the challenges posed by grouped distributions, experiencing a notable decline in performance by 20.3% and 22.3%. This observed decrease in performance is consistent across models, unaffected by the specific positions of the labels within the prompt.

The potential reasons why there is a substantial drop in performance after grouping the labels in demonstrations are worthy of further research to explore and offer valuable insights for future research and development in this area. One of the potential explanations is that Large Language Models can develop biases based on the position of examples within the prompt. Grouping similar examples together may accidentally reinforce such biases, leading the model to overfit to specific patterns associated with certain positions (like the end or the beginning of the demonstration context). Scattering demonstrations helps distribute examples more evenly, reducing the likelihood of position-induced biases as discussed in Lost-in-the-Middle (Liu et al., 2023).

## 5    Conclusion

In summary, our research explores the capability of LLMs on long in-context learning tasks, particularly in extreme-label classification scenarios. We curate a dataset *LongICLBench* consisting of long in-context learning tasks with different difficulty levels in terms of context length. Through our study, we have discovered that LLMs demonstrate dramatic performance degradation when it comes to more difficult tasks. Our exploratory experiments further highlight the impact of the distribution of examples within prompts on model performance. We hope *LongICLBench* and our findings contribute to the ongoing efforts to enhance LLMs' understanding of long contexts.

### Broader Impact Statement

The development of long LLMs evaluation benchmarks and the corresponding insights can boost the development of long-context techniques, which can revolutionize fields requiring deep contextual understanding, such as legal analysis, long-form journalistic content generation, and comprehensive academic summarization. However, there are potential risks associated with the deployment of such powerful models. Enhanced long-context capabilities could be misused for generating misinformation, especially in political or social contexts, where nuanced long-form content can have significant influence. There is also the risk of dependency on automated systems in critical decision-making processes, which could lead to over-reliance on technology at the expense of human judgment.

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

## A  Appendix

### A.1  Additional Datasets

We list a few additional datasets as follows:

**GoEmotions** (Demszky et al., 2020) is the largest manually annotated dataset of 58k English comments from Reddit, which is labeled into 27 emotion categories or Neutral. There are 27 types of emotion types and drop the rare ones with few examples. Each selected example contains 28 tokens on average.

**Few-NERD** (Ding et al., 2021) is a large-scale human-annotated name entity recognition dataset with a hierarchy of 8 coarse-grained and 66 fine-grained entity types. Each of the instances is a paragraph with approximately 61 tokens on average and contains one or multiple entity names as the ground truth answer. There are 66 types of entities in the collection.

The performance for the two tasks is demonstrated in Table 7 and Table 8.

| Model | Param | Support | 1R | 2R | 3R | 4R | 5R |
|---|---|---|---|---|---|---|---|
| **Context Tokens** | | | **0.8K** | **1.6K** | **2.4K** | **3.2K** | **4K** |
| Gemma-7B-base | 7B | 8K | 0 | 0 | 0 | 0 | 0 |
| LLaMA-2-7B-32K | 7B | 32K | 0 | 0 | 0 | 0.2 | 0.2 |
| ChatGLM3-6B-32K | 6B | 32K | **22.0** | 17.0 | 15.0 | 12.6 | 10.6 |
| Qwen-1.5-7B-base | 7B | 32K | 14.8 | **18.2** | **18.6** | **19.0** | **14.2** |
| Mistral-7B-v0.2-base | 7B | 32K | 2.6 | 11.4 | 7.4 | 11.6 | 12.4 |
| LLaMA-2-7B-LongLora | 7B | 100K | 0 | 0 | 0 | 0 | 0 |
| Yi-6B-200K | 6B | 200K | 0 | 0 | 0.8 | 4.0 | 4.0 |
| InternLM2-7B-base | 7B | 200K | 0 | 0 | 0 | 0 | 0 |
| Long-LLaMA-code-7B | 7B | 256K | 0 | 0 | 0 | 0.2 | 0.4 |
| RWKV-5-World | 7B | 4K | 8.8 | 7.4 | 4.6 | 5.2 | 4.0 |
| Mamba-2.8B | 2.8B | 2K | 0 | 0 | 0 | 0 | 0 |
| GPT4-turbo | N/A | 128K | **36.5** | **34.4** | **35.0** | **33.3** | **32.0** |
| GPT4o | N/A | 128K | 23.0 | 23.8 | 21.2 | 21.2 | 22.2 |
| Claude3-Opus | N/A | 200K | 25.8 | 7.4 | 17.0 | 12.6 | 19.6 |
| Gemini-1.5-Pro | N/A | 10M | 19.0 | 10.4 | 9.2 | 10.6 | 9.4 |
| SoTA (BERT) | N/A | - | | | **58.9** | | |

Table 7: GoEmotion Result.

| Model | Param | Support | 1R | 2R | 3R | 4R | 5R |
|---|---|---|---|---|---|---|---|
| **Context Tokens** | | | **5K** | **9K** | **14K** | **19K** | **24K** |
| Gemma-7B-base | 7B | 8k | **44.0** | 44.2 | 0 | 0 | 0 |
| LLaMA-2-7B-32K | 7B | 32k | 36.9 | 40.8 | 41.1 | 41.6 | 41.3 |
| ChatGLM3-6B-32K | 6B | 32k | 24.1 | 9.3 | 23.6 | 10.4 | 1.1 |
| Qwen-1.5-7B-base | 7B | 32k | 40.0 | 46.4 | 47.6 | 47.3 | 47.8 |
| Mistral-7B-v0.2-base | 7B | 32K | 42.2 | **47.4** | **48.9** | **50.0** | **50.0** |
| LLaMA-2-7B-LongLora | 7B | 100K | 0 | 0 | 0 | 0 | 0 |
| Yi-6B-200K | 6B | 200k | 34.3 | 40.2 | 44.8 | 42.3 | 43.2 |
| InternLM2-7B-base | 7B | 200k | 43.6 | 46.2 | 46.5 | 47.8 | 48.3 |
| Long-LLaMA-code-7B | 7B | 256K | 22.3 | 25.5 | 26.5 | 29.4 | 27.0 |
| RWKV-5-World | 7B | 1k | 13.9 | 0 | 0 | 0.7 | 9.9 |
| Mamba-2.8B | 2.8B | 2k | 0 | 0 | 0 | 0 | 0 |
| GPT4-turbo | N/A | 128k | 53.4 | **55.3** | **56.2** | **55.6** | **56.8** |
| GPT4o | N/A | 128k | 46.7 | 41.4 | 42.8 | 39.0 | 44.4 |
| Claude3-Opus | N/A | 200k | 53.5 | 51.3 | 51.2 | 52.4 | 52.5 |
| Gemini-1.5-Pro | N/A | 10M | **55.4** | 47.8 | 49.5 | 41.4 | 42.4 |
| SoTA (PL-Marker) | N/A | - | | | **70.9** | | |

Table 8: Few-NERD Result.

## A.2 Prompting Template

The prompting template for each of the datasets is presented at Table 9

## A.3 Additional Distribution Analysis

To facilitate a clear comparison between random and grouped distributions, we organize instances of the same class to be adjacent within the demonstration prompts. The impact of this reorganization on model performance, both pre and post-grouping, is presented in Table 10.

| Dataset | Prompt |
|---|---|
| GoEmotion | Given a comment, please predict the emotion category of this comment. The prediction answer must come from the demonstration examples with the exact format. The examples are as follows:
{comment: "...comment..."
emotion category: "...emotion..."
} × *repeat n times* |
| BANKING77 | Given a customer service query, please predict the intent of the query. The predicted answer must come from the demonstration examples with the exact format. The examples are as follows:
{service query: "...service..."
intent category: "...intent..."
} × *repeat n times* |
| TacRED | Given a sentence and a pair of subject and object entities within the sentence, please predict the relation between the given entities. The examples are as follows:
{sentence: "...sentence...
the subject is "...subject..."
the object is "...object..."
the relation between the two entities is: "...relation..."
} × *repeat n times* |
| Few-NERD | Given the sentence, please find the name entities in the sentence and their corresponding entity types in the strict format of the given examples as following (Entity: EntityType):
{"...entity...": "...entity type..."
} × *repeat n times* |
| DialogRE | Given the dialogue, please find the name pair entities in the dialogue and their corresponding relation types in the strict format of given examples as following (note that the number of entities has to strictly have the same value as the number of respective relation):
{Dialogue:
"...dialogue..."
The list of entity pairs are "...(subject1, object1), (subject2, object2), etc...
The "...number of pairs..." respective relations between each entity pair are: "...relation, relation2, etc...
} × *repeat n times* |
| Discovery | Given two sentence1 and sentence2, please predict the conjunction word between the two sentences. The predicted answer must come from the demonstration examples with the exact format. The examples are as follows:
{"...sentence1..." ( ) "...sentence2..."
the conjunction word in ( ) is "...conjunction..."
} × *repeat n times* |

Table 9: The data prompt format of each dataset. Each dataset has a unique prompt format to effectively utilize the context and format of its respective data to get the best output response.

The distribution plots for other models are shown in Figure 5 and Figure 6.

## A.4  Data Accessibility

Our LongICLBench is set under MIT license, thus permission is granted, free of charge, to any person obtaining a copy of this dataset and associated documentation files. The datasests are curated under the rules guaranteed by the original dataset. There is no personally identifiable or offensive content in the dataset.

| Model | Param | Support | Scatter | Grouped | Δ |
|---|---|---|---|---|---|
| **Context Tokens** | | | | **10K** | |
| Gemma-7B-base | 7B | 8K | 0 | 0 | 0 |
| LLaMA-2-7B-32K | 7B | 32K | 0.4 | 3.0 | +2.6 |
| ChatGLM3-6B-32K | 6B | 32K | 38.9 | 35.6 | -3.3 |
| Qwen-1.5-7B-base | 7B | 32K | 45.2 | 33.0 | -12.2 |
| Mistral-7B-v0.2-base | 7B | 32K | 51.6 | 5.1 | -46.5 |
| LLaMA-2-7B-LongLora | 7B | 100K | 0 | 0 | 0 |
| Yi-6B-200K | 6B | 200K | 8.0 | 0 | -8 |
| InternLM2-7B-base | 7B | 200K | 15.5 | 4.8 | -9.7 |
| Long-LLaMA-code-7B | 7B | 256K | 4.1 | 0 | -4.1 |
| RWKV-5-World | 7B | 4K | 1.0 | 3.6 | +2.6 |
| Mamba-2.8B | 2.8B | 2K | 0 | 0 | 0 |
| GPT4-turbo | N/A | 128K | 79.5 | 59.2 | -20.3 |
| Gemini-1.5-Pro | N/A | 10M | 79.6 | 57.3 | -22.3 |

Table 10: Exploratory Result on TacRED 3 Round. **Grouped** means forcing the same-typed demonstration examples near by each other instead of randomly distributing in the prompt.

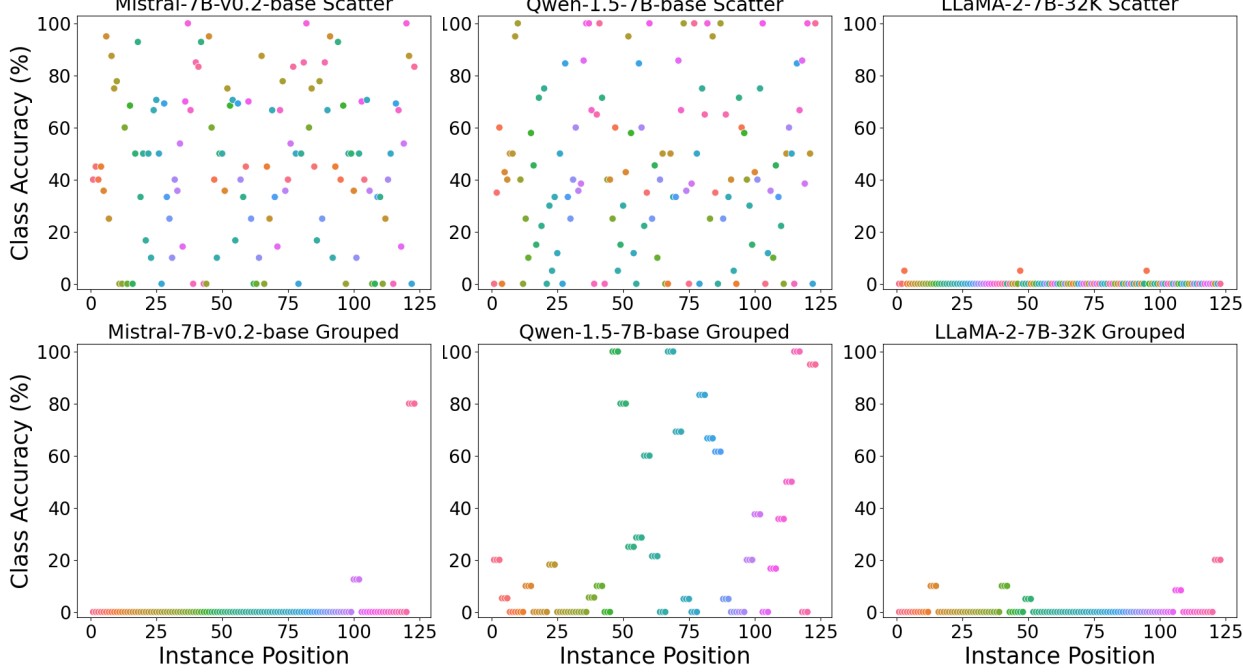

Figure 5: Visualization of accuracy for every class when instances from the same class are scattered V.S. grouped in the demonstration prompt.

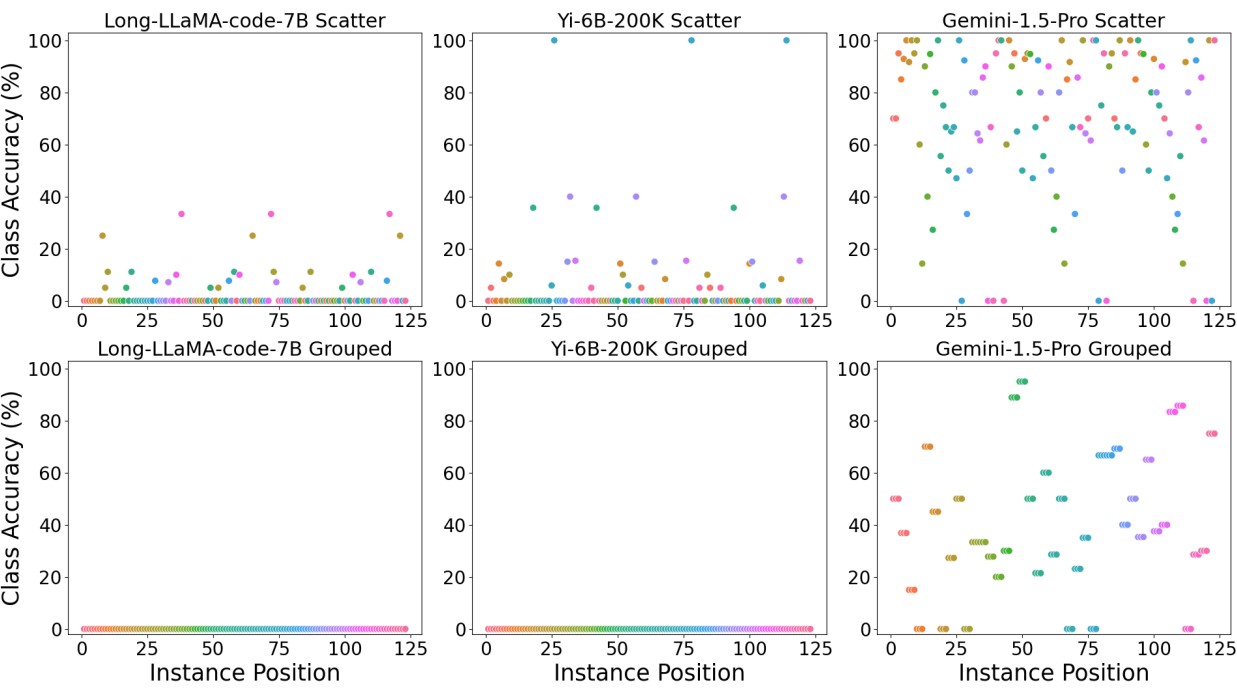

Figure 6: Visualization of accuracy for every class when instances from the same class are scattered V.S. grouped in the demonstration prompt.

