# OpenReview forum: "Long-context LLMs Struggle with Long In-context Learning"
_TMLR — Accepted by TMLR_

### Review · Reviewer_iSUr · 2024-10-06

**Summary Of Contributions:**

This paper provides an experimental evaluation on the performance of LLMs processing long sequences.
The current evaluation for LLMs replies on basic metrics like perplexity and synthetic tasks.
However, these metrics may not be able capture the real world scenarios.
Indeed, previous results showed that LLMs handle shorter and simpler tasks well but they struggle on more difficult tasks.

In this paper, the authors introduce a new benchmark called LongICLBench which focuses on long in-context learning for extreme-label classification.
The result suggests that there are some areas needed to be improved.
The first one is Bias towards later labels in the sequence.
The second one is performance degradation as input length and label space increase.
The third one is the need for better reasoning over long contexts and multiple pieces of information.

**Audience:**

Yes

**Broader Impact Concerns:**

.

**Claims And Evidence:**

Yes

**Requested Changes:**

Note:

- Abstract: "Some models like Gemini could even to be capable ..." $\to$ "Some models like Gemini could even be capable ..."

**Strengths And Weaknesses:**

Strengths:

- This paper introduces a novel benchmark, LongICLBench, which will undoubtedly aid in the future development of LLMs by assessing their performance on long sequences.
The experimental studies effectively highlight the shortcomings of current LLMs.



Weaknesses:

- Although the paper points out the issues with current LLMs, it does not provide concrete solutions to address them.
As such, this paper serves primarily as a diagnostic tool.
That being said, it may have limited impact on researchers looking to propose solutions.

---

> ### Author Response · Authors · 2024-11-18
>
> We thank the reviewer for the feedback and for acknowledging the strengths of our work. The core contribution of this paper lies in identifying the limitations of current long-context evaluation benchmarks, which often rely on metrics like perplexity or synthetic tasks that fail to reflect real-world challenges. To address this, we introduce LongICLBench, which leverages extreme-label classification in an in-context learning setting as a natural testbed for evaluating the true long-context capabilities of LLMs. This approach requires models to process and reason over extensive inputs while navigating large label spaces, providing a more realistic measure of their performance. Through our experiments, we highlight critical limitations of existing models, including bias toward later labels, performance degradation with increasing input length and label space, and challenges in reasoning over long contexts.
>
> Regarding the observation of lack of proposed solution, while there exists a wide range of solutions for long-context models, these methods often report highly divergent observations, which we believe are influenced by evaluation biases inherent in existing benchmarks. To address this, we designed LongICLBench to provide a more comprehensive and unbiased evaluation. Our benchmark covers models trained with a variety of long-context methods, enabling us to reveal consistent patterns and limitations across diverse approaches. These findings offer actionable insights for future research to explore mitigation strategies and improve long-context capabilities. This diagnostic perspective is the primary goal of our paper.
>
> We also appreciate the reviewer pointing out the typo in the abstract, and we have corrected it in the revision.

---

### Review · Reviewer_7dKU · 2024-10-12

**Summary Of Contributions:**

This paper proposes LongICLBench, a new benchmark for evaluating long-context LLMs. The core idea is to use extreme-label classification datasets to construct the benchmark. The authors compare 15 long-context LLMs across 6 datasets and suggest that modern long-context LLMs still struggle with these tasks.

**Audience:**

Yes

**Broader Impact Concerns:**

I have no concerns about the ethical implications of the work beyond those stated in the paper.

**Claims And Evidence:**

Yes

**Requested Changes:**

There is no critical adjustment required. Along with the weaknesses mentioned above, here is an additional change that could strengthen the work.

1. Can the dataset be easily extended to include more demonstrations (e.g., 10-shot)? Having the flexibility to provide longer inputs would be beneficial for future use of the benchmark, as most LLMs are already claiming context windows of 128k tokens or even millions.

**Strengths And Weaknesses:**

Strengths
1. The paper proposes a novel and convincing approach for evaluating the long in-context learning capabilities of LLMs.
2. The authors conducted extensive evaluations across diverse models.
3. The paper is clearly written and easy to follow.

Weaknesses
1. There is no explanation for how the specific datasets were selected. Can the same approach be directly applied to other extreme-label classification datasets to extend the benchmark?
2. The authors demonstrate that scattering the demonstrations consistently performs better than grouping them, but they do not provide an explanation for the suspected reasons.

---

> ### Author Response · Authors · 2024-11-18
>
> We thank the reviewer for pointing out the need for clarifications on dataset selection and the suggestion to extend the benchmark to longer contexts. Below, we address these concerns in detail.
>
> **Clarification of Data Selection**
>
> Regarding dataset selection, the datasets were chosen based on their suitability for extreme-label classification tasks, with a focus on those containing a substantial number of label types (at least more than 20 classes). This criterion ensures that multiple rounds of traversing the label types can naturally span the context length to long-context scenarios. Moreover, we selected datasets with increasing difficulty levels, in this most challenging case, a single round of traversing all label types results in a context length of 10K tokens, revealing significant limitations of current models. As demonstrated in our evaluation, most existing long-context models fail to perform satisfactorily in this setting. Moreover, the same approach can be directly applied to other extreme-label classification datasets to extend the benchmark, and we describe the conversion process in Section 3.1.
>
> **Extension of the Benchmark to Longer Contexts**
>
> We initially limited the evaluation to a maximum of 5-shot demonstrations based on our findings. Specifically, we observed that most open-source models began to degrade in performance after 3 rounds of demonstrations (e.g., in simpler datasets like BANKING77 and DialogueRE) and completely failed in the hardest dataset, Discovery, where context lengths reached around 10K tokens with just one-round of demonstration. This indicates the current long-context LLM’s deficiency in handling even moderate context lengths (7K–20K), falling significantly short of the claimed capacities (32K). Moreover, in the most difficult settings, even API-based models failed when the context length approached 50K tokens. We appreciate the reviewer’s suggestion and agree that extending the benchmark to include longer contexts is necessary. Specifically, we will increase the Discovery dataset’s demonstrations to span up to 128K tokens. This enhancement will provide a more rigorous testbed for evaluating the potential of future long-context models. We will link the extended dataset to this paper once it is deanonymized.
>
> **Explanation on Scattering V.S. Grouping**
>
> On the observation that scattered demonstrations perform better than grouped ones, we think the explanation is that Large Language Models can develop biases based on the position of examples within the prompt. Grouping similar examples together may accidentally reinforce such biases, leading the model to overfit to specific patterns associated with certain positions (like the end or the beginning of the demonstration context). Scattering demonstrations helps distribute examples more evenly, reducing the likelihood of position-induced biases as discussed in Lost in the Middle [1]. We have included a brief discussion of this point in the revised paper.
>
> [1] Liu, N.F., Lin, K., Hewitt, J., Paranjape, A., Bevilacqua, M., Petroni, F., & Liang, P. (2023). Lost in the Middle: How Language Models Use Long Contexts. *Transactions of the Association for Computational Linguistics, 12*, 157-173.

---

### Review · Reviewer_4FTQ · 2025-01-02

**Summary Of Contributions:**

The authors offer a new benchmark dataset, LongICLBench, for long in-context learning in extreme-label classification. This dataset comprises six tasks with different difficult levels regarding context length and label space. Using this dataset, this paper exhibits the performance of numerous long-context LLMs and demonstrates that the models'  performance consistently degrades as given tasks get more complex.

**Audience:**

Yes

**Claims And Evidence:**

Yes

**Requested Changes:**

1. Regarding the two weaknesses above, could the author add more information to the paper?
2. It would be helpful to report the performance achieved by employing other learning techniques with the entire dataset, not using ICL, or human evaluation. This would provide an oracle performance level, offering clearer context for how underwhelming the ICL performance truly is.

**Strengths And Weaknesses:**

Strengths
* The introduction of the new benchmark in this paper could contribute to evaluating long-context capabilities in LLM for more practical scenarios focusing on tasks that require understanding large contexts and extreme-label classification.
* The authors provide some insights which are the degradation of performance with increased context length and potential bias affecting prediction outcomes.

Weaknesses
1. This paper lacks an in-depth explanation of why LLMs fail to generalize with longer contexts or extreme labels. It just reported that the length of contexts affects the performance degradation.
2. This paper lacks specific explanations for their datasets. The author should contain more concrete information about the datasets such imbalance of the dataset or what the classification labels are. Furthermore, since this paper makes ICL scenarios with existing benchmark datasets, it also should note how differently those existing datasets were used.
3.

---

> ### Author Response · Authors · 2025-01-13
> **Comment**
>
> Thank you for your insightful feedback on our work. Below, we address the concerns you raised:
>
> ### Lack of in-depth explanation of performance degradation
>
> We acknowledge the need for further discussion on why LLMs fail to generalize effectively with longer contexts or extreme labels. Our study hypothesizes that the degradation arises primarily from two factors: (1) the models’ inherent inability to manage the exponentially increasing complexity of longer contexts (larger label space and longer token sequence), and (2) biases introduced by positional dependencies within demonstrations, as evidenced by our "grouped vs. scattered distribution" experiments. While these points are mentioned in Sections 4.1 and 4.2, we will include a more detailed explanation in the revised manuscript to make these factors clearer.
>
> ### Insufficient description of datasets
>
> Regarding the concern about dataset descriptions, we clarify that the sub-datasets used in LongICLBench have a balanced label distribution. This is achieved by evenly sampling instances across all classes, as detailed in Section 3.1. To address the reviewer’s request for more concrete information, we will expand the appendix to include:
>
> 1. A detailed description of the label space for each dataset.
> 2. Additional clarification on how existing benchmark datasets were adapted for the LongICLBench scenarios, including any modifications made.
>
> ### Including performance from other learning techniques
>
> We appreciate the suggestion to compare the ICL performance with results achieved through alternative techniques or human evaluation. We have already included state-of-the-art (SoTA) results for each dataset in the results tables (e.g., Table 6 for Discovery). These results demonstrate a significant performance gap between ICL and fully trained SoTA models, highlighting the challenges posed by long-context ICL approaches.
>
> We hope these additions will address the raised concerns and improve the clarity and completeness of our work.

---

### Decision · Action_Editor_vJaf · 2025-02-19

**Recommendation:** Accept as is

**Comment:**

The paper introduces LongICLBench, a new benchmark that leverages extreme-label classification tasks to evaluate the long-context capabilities of Large Language Models (LLMs). The benchmark provides a more realistic assessment compared to traditional metrics like perplexity, demonstrating through extensive experiments across multiple models and datasets that current models struggle with longer contexts and larger label spaces.

The reviewers unanimously agree that the benchmark's design and comprehensive evaluation make a valuable contribution to the field. The authors have adequately addressed the reviewers' concerns in their rebuttal, including clarifications on dataset selection criteria, the impact of demonstration ordering, and detailed dataset descriptions. While one reviewer noted the lack of proposed solutions, the Action Editor agrees that establishing a robust diagnostic benchmark is an important contribution that will enable future research on improving long-context capabilities.

**Audience:**

Yes

**Claims And Evidence:**

Yes